# Loss of years of healthy life due to road incidents of motorcyclists in the city of Medellin, 2012 to 2015

**Sandra Milena Porras Cataño[1], Hugo Grisales-Romero[2]***

**1** University of Antioquia, Medellin, Colombia, **2** National School of Public Health, University of Antioquia, Medellin, Antioquia, Colombia

* hugo.grisales@udea.edu.co

## Abstract

### Objective

Determine the loss of years of healthy life due to road incidents of motorcyclists in the city of Medellin from 2012 to 2015.

### Methods

Descriptive study with data on health care of injured motorcyclists and deaths adjusted with the Preston and Coale method, and OPS proportional distribution for the period 2012–2015. The years of life lost due to premature death (YLLs), years lived with disability (YLDs), and the disability-adjusted life years (DALYs) were calculated according to the new methodology designed for that purpose.

### Results

The loss of years of healthy life due to road incidents of motorcyclists in the four-year period was 80,046 DALYs (823.8 per 100,000 inhabitants), with a higher proportion in men (81.3% and a ratio of 5 to 1 compared to women); the YLDs was 66.6% with marked differences in favor of men. There was nearly a 38% difference in the ages of 15 to 19 as well as a 19% difference from 30 to 49, compared to women. Premature death (YLLs) contributed to 33.4% of DALYs, with significant presentation in the above-mentioned age groups.

### Conclusions

The greatest loss of years of healthy life due to road incidents of motorcyclists in Medellin was due to non-fatal injuries and was concentrated in young men. If the trend of motorcycle road incidents continues, both local and national road safety plans will fail to accomplish the expected results, especially among motorcycle users.

**Data Availability Statement:** All seven (7) files are available from the Zenodo database (accession number 10.5281/zenodo.4836303 / URL: https://zenodo.org/record/4836304#.YLEcJ6hKguU).

**Funding:** Financed by Colciencias Science, Technology, and Innovation Program (Ciencia, Tecnología e Innovación) through Call No. 744-2016, Contract No. 633-2017 and Research Group on Demography and Health, National Faculty of Public Health, University of Antioquia, Medellin, Colombia.

**Competing interests:** The authors have declared that no competing interests exist.

## Introduction

An accident is defined as a random event that occurs by chance and disrupts the normal order of things. In this event, uncontrollable causes act that can affect various actors, and in which circumstances that cannot be foreseen or avoided. In contrast, traffic incidents have a causal origin, if a driver violates traffic regulations and causes a fatal event, that fact is causal, it is not subject to chance, and therefore it is not an accident. A preventable event is not probabilistic [1].

Injuries have been incorrectly known as accidents. This misconception was strengthened because injuries were not considered as priority events in public health. According to the World Health Organization (WHO), injuries have specific causes, are predictable, preventable, and have repetitive patterns, just like violence-related injuries in all settings [1]. Current approaches to injury prevention require that the term "accident" be replaced by "injury" or "event" or "incident." If the conceptual error of considering preventable events as "accidents" is understood, the objective of reducing the incidence of injuries and deaths in traffic events could be achieved.

In Colombia, both terms are not adequately differentiated, and a paradigm shift is urgently required [2]. In this context, the term "traffic accident" is identified as "road incident" [1] because it is the result of inappropriate behavior of people, such as speeding, driving under the influence of alcohol, not wearing a helmet or seat belt, among others [3, 4]. It can also be explained by exogenous factors such as the increase in the number of vehicles, population growth and inadequate urbanization that make the infrastructural capacity for road regulation insufficient [5].

There are other endogenous factors to be taken into account in RIs, especially the human factor. In Colombia, Norza et al. in 2014, in a sample of 16,322 drivers, reported the role of motorcyclists in RIs, probably attributed to the disregard of traffic regulations, which added the fact of the lack of experience behind the wheel that characterizes these people. The long working hours combined with the drivers' lack of emotional intelligence could be triggers of stress, anxiety, anger, aggressiveness, hostility, dissociative disorders, and consequently, factors that facilitate the tendency to commit more infractions and RIs [6]. Also, in three Colombian cities, Ibague-Valledupar, and Bogotá, two studies ratified the precept of the multi-causality of traffic accidents, where human factors are the ones that most contributed to the occurrence of incidents, aggravated by apparent deficiencies in infrastructure and some organizational problems [7, 8].

The annual report Traffic Accidents in the Andean Community 2007–2016 evidenced a gradual increase in road incidents (hereinafter RIs) from 318,406 in 2010 to 379,348 in 2015. Minor motor vehicles, within which motorcycles are included, ranked second among vehicles involved in RIs at 22.7% [9].

RIs are a public health problem. According to the WHO, projections place them as the fifth leading cause of death by the year 2030 [3, 4]. In fact, in 2016, it was revealed that 1.35 million people died in the world due to RIs, at least 10 million were seriously injured, and 25 million more were moderately injured; 50 million more suffered mild effects, but most of these events have not been recorded [3, 4, 10]. RIs were the eighth leading cause of death in all age groups and the first among children and young adults between 5 and 29 years old [4], where the most vulnerable users of public roads were motorcyclists, pedestrians, and cyclists (at 28%, 23%, and 3% respectively), with disparities by region and Gross Domestic Product (GDP) of the countries [4].

Motorcycles are one of the most important goods in Colombian households; as revealed by the Unique National Traffic Registry (RUNT, by its Spanish acronym), in 2016, there were

more than 7.2 million motorcycles in the country [10]. The increase in this means of transport has placed a significant burden on the health of the population, which is manifested in traumas, injuries and deaths resulting from accidents. Of the total number of drivers involved in traffic accidents, motorcyclists contributed 77.84% of the deaths and 78.78% of the injured; and regarding passengers, they have contributed 50.13% of the deaths and 50.17% of the injured [11]. The impact on the quality of life and on the health system due to the consequences produced by RIs of motorcyclists, as well as the high economic costs that they entail, warrants study.

RIs in the department of Antioquia have grown almost exponentially; consequently, both deaths and injuries have increased with a predominance in the city of Medellin [11, 12]. Towards 2014, in its annual report on accidents, the Medellin Mayor's Office revealed that motorcycle users (drivers and passengers) and pedestrians were the most vulnerable road actors in the city, with 43.4% and 47.6% of traffic-related deaths; it was also discovered that young people between 20 and 29 years of age constituted 41.6% of the motorcyclists who died. Motorcycles were involved in approximately 55% of the accidents present in the city and in 44% of the deaths of pedestrians [13].

In Medellin, the problem concerning RIs of motorcyclists has been addressed in morbidity and mortality studies, with simple indicators, analyzed independently, which prevent an overall perspective of the burden of the disease. In that light, in order to attain a more holistic vision in accordance with the guidelines of the World Bank (WB), the WHO, and the Institute of Health Metrics and Evaluation (IHME), an initiative was proposed to construct a single indicator that reflects the losses due to premature mortality (years of life lost –YLLs–) as well as disability (years lived with disability –YLDs–). The goal is for these losses to converge at the estimation of disability-adjusted life years (DALYs) due to RIs of motorcyclists in the city of Medellin between 2012 and 2015. This way, the city's transportation policy makers will have information to monitor the magnitude, severity, and burden of RIs, thus supporting the prioritization, improvement, adaptation, and adoption of new measures that strengthen their intervention and control in a systemic and specific way for everyone on the road.

## Materials and methods

### Study type

This is a descriptive study of the burden of the disease, with secondary information sources, the methodology which adopts the latest variants related to the estimation of DALYs as it is detailed in the scientific literature [14, 15].

### Study population

These were all records of deaths and injuries due from RIs of motorcyclists, collected from different sources, in the city of Medellin between 2012 and 2015.

### Units of analysis

Three units were considered, namely: each death associated with the external cause corresponding to RIs of motorcyclists for both drivers and passengers (according to the International Classification of Diseases of the WHO, in its tenth revision -ICD 10-, from code V20 to V29); each motorcyclist (driver or passenger) who was injured at the scene of the events according to the severity recorded in the Police Reports of Traffic Accident (IPAT, for the Spanish acronym) and each health care to the injured motorcyclist with at least one of the injury types considered in the analysis [15, 16] (S1 Table).

### Information sources

The sources of information were analyzed by the unit of analysis and processed independently due to the absence of a tracer identifier. For the characterization of deaths and the calculation of YLL, the database of the Unique Affiliate Registry Death Module (RUAF-D, for the Spanish acronym) was used, which was provided by Department of Antioquia's Local Bureau of Health and Social Protection (SSSA, for the Spanish acronym) (S1 Dataset). For the knowledge of the distribution of non-fatal injuries of motorcyclists, we used the injured people database according to IPAT at the scene of the events—"in situ", provided by the Secretary of Mobility of Medellin (S2 Dataset). For the estimation of YLD, we used the consolidated data of the Secretary of Health of Medellin (Secretaría de Salud de Medellín) from the Individual Records of the Provision of Health Services (RIPS, for the Spanish acronym) of hospitalization, outpatient consultation and emergency services with previous validation; only information on morbidity consolidated by the territorial entity between 2012 and 2015 was collected, which limited the study of the burden of disease to that four-year period (S3 Dataset).

### Inclusion criteria

All deaths and injuries from RIs of motorcyclists (drivers and passengers who required attention in outpatient, hospitalization or emergency health services) reported in Medellin between the years 2012 and 2015.

### Exclusion criteria

All deaths due to RIs in which the motorcyclist status was not specified; also duplicated records of events and injured with type of injury due to RIs of motorcyclists that did not have the nature of the injury defined were excluded [15, 16] (S1 Table).

### Data processing

For data standardization and data mining, the Cross Industry Standard Process for Data Mining (CRISP-DM) methodology was adopted [17]. The first phase consisted of data collection and identification of quality problems (Data understanding); the second, focused on their transformation, cleaning, and integration; both procedures were carried out with specific extraction, transformation (cleaning), and loading techniques. Duplication, completeness, content, and consistency problems were corrected until the quality of the information was optimized. The underreporting of mortality that occurred in Medellin due to RIs was estimated and adjusted according to the Preston and Coale [18] method, which equated to 1.2%.

### Bias control

The feasible selection and information biases were controlled by evaluating the completeness and relevance of the information on the outcomes originated by the RIs of motorcyclists, according to the data source to be used and the diagnostic coding to be taken into account in accordance with the International Classification of Diseases, 10th Revision (V20-V29) and the assessment of the coverage of the underreporting of mortality in addition to the possible errors in the coding of healthcare attentions.

### Statistical analysis

Initially, an analysis was made on the deaths and injuries of motorcyclists for the period 2012–2015, according to the following variables: year, sex, age, age group, marital status (available only in the source for deaths), class and time of RI of motorcyclists and condition.

For the calculation of the two components of DALYs, YLLs, and YLDs, the methodology used in the Global Burden of Disease (GBD) studies was followed, adopting the latest theoretical novelty from 2010. For the calculation of the YLLs, the method of standard expected life years lost was used, with which deaths at all ages, even after the standard, contribute to the total estimated burden of disease. The frontier national life expectancy projected for the year 2050 of 91.9 years for both men and women was considered a standard in order to represent the maximum useful life of a person in good health who is not exposed to avoidable risks or serious injuries, and it allows comparison between different countries. First, the YLLs were calculated by differentiating the class mark of the age in each five-year group and the standard national life expectancy; second, by multiplying the differences found by the number of deaths in each age group; and third, by adding the previous volumes.

For the estimation of YLDs, according to the last proposed methodological variant, the discount rate and the age weights were omitted. First, it was necessary to identify the events with the type of injury (cause of harm) "RIs of motorcyclist" (ICD 10-V20-V29) by nature of injury (bodily harm) for each year, age group, and sex. To do this, the variables of external cause and main and related diagnoses of the RIPS were taken into account. To avoid overestimating the cases, the care related to each patient was followed; and when the patient had care registries for several health services within a very short time span, a prioritization was made by service as follows: hospitalization, outpatient, then emergency. Multiple registries with the same diagnosis and more than 90 days between the first admission and subsequent admissions were identified as independent cases, as considered in other studies. It was necessary to estimate the number of times a person received medical attention for a given motorcycle-related RI. As the means of transportation was occasionally omitted from RI reports, the estimation was gathered with the use of interdependence techniques such as the categorical principal components analysis and discriminant analysis and establishing the type of injury as the outcome.

Next, the number of cases of RIs of motorcyclists treated by nature of the injury were divided among the population of Medellin according to year, sex, and age group. This was done in order to identify the proportion of individuals that were affected by any of the natures of injury of interest due to a RI of motorcyclist in each of the years considered, as it was the epidemiological parameter used. Likewise, according to the nature of the short-term and untreated injuries, the disability weights defined by the WHO in the GBD 2013 were used [15, 16, 19] (S2 Table).

An adjustment according to the multiplicative model was necessary due to the fact that the YLDs could be overestimated if a significant volume of individuals presented multiple injuries due to the same incident. Furthermore, if the loss of health for each injury were to be counted separately, it would reflect considerably inflated results.

The natures of injury were ordered in descending order according to the disability weights and their adjusted weights (Eq 1) and the adjusted epidemiological parameter were calculated (Eq 2), as indicated below:

Eq 1

$$DW_{1+2} = 1 - (1 - DW_1) * (1 - DW_2) \tag{1}$$

Where $DW_{1+2}$ is the weight of disability weighted by more than one injury that a person suffers, $DW_1$ is the weight of disability assigned to the first injury, and $DW_2$ is the weight of disability assigned to the second injury.

Eq 2

$$p_{1+2} = p_1 + p_2 - p_1 * p_2 = (1 - (1 - p_1) * (1 - p_2)) \tag{2}$$

Where $p_{1+2}$ is the prevalence of the two comorbid Injuries 1 and 2, $p_1$ is the prevalence of Injury 1 and $p_2$ is the prevalence of Injury 2.

Then, to calculate the YLDs adjusted by the multiple injuries that a person could suffer, the disability weights and the epidemiological parameters given by Eqs 1 and 2 were multiplied. Thus, the total YLDs of motorcyclists per year of study corresponded to the sum by age group and sex of YLDs. Accepting the possible combination of natures of injuries was unavoidable."

To obtain the DALYs, the YLLs and YLDs were summed up per age group and sex for each of the years individually and the four-year period as a whole. For their description, absolute and relative frequencies, rates per 100,000 inhabitants, and the 95% uncertainty intervals are presented, according to the Bootstrap resampling technique.

### Ethical considerations

The research was considered without risk according to Resolution 8430 of 1993, Article 11, numeral a), of the Ministry of Health of Colombia [20].

### Information processing

For the application of the technical strategies and the results, the PostgreSQL database manager, the statistical software IBM SPSS 21Ⓡ and the Microsoft Excel spreadsheet software were used.

## Results

### Overview of mortality and injuries due to RIs of motorcyclists

In Medellin between 2012 and 2015, there were 428 deaths of motorcyclists due to RIs, most of them unmarried, 53,7%; 93.2% of the deceased were between 15 and 49 years old, more frequently between 15 and 29 years old (54.6% and average mortality rate of 10.1 per one hundred thousand); the deceased were also mostly men (84.8%), of whom 86.0% were drivers. The women who died as either the driver or a passenger constituted 5.6% and 8.2%, respectively of the total. On average, there were 107 deaths per year, the greater percentage being in 2013 (26.6%). The average mortality rate was 4.4 per 100,000 inhabitants, higher for men (7.9 per 100,000 for men to just 1.3 for women (Table 1).

Regarding the 87,971 motorcyclists injured during the period, of which 79.5% were drivers, mainly men (73.7%) and in 50% of the cases they were 27 years old or younger (IQR: 13); only in 13.0% and 13.3% of the cases, women were drivers or passengers, respectively. Each year, an average of 21,993 (95% UI: 17,417–24,404) motorcyclists were injured; the rate of injury was unstable, with an upward trend, from 822.7 to 979.6 per 100,000 inhabitants in 2012 and 2015, respectively; the average rate of injury was 904.9 (95% UI: 726–990) per 100,000 inhabitants, and it was men who had the greatest contribution to non-fatal injuries in the study period; crashes with other vehicles, resulting in motorcyclists killed and injured, were predominant in the study period (Table 1, S3 Table). The peak times for motorcyclist road incidents, with fatalities, were between 4:00 a.m. and 8:00 am. (21.2%) and from 8:00 p.m. to midnight (20.6%), while for the injured, from 6:00 a.m. to 8:00 a.m. (13.9%) and from 4:00 p.m. to 8:00 p.m. (23.3%).

It is important to clarify that of the 87,971 injured persons mentioned above-whose information was obtained from the official "in situ" reports of the Secretary of Mobility of Medellin-not necessarily all required medical attention or hospitalization; however, with the source of information of the RIPS, which reports the injuries that required attention in a health service, it was found that in the four-year period, there were 45,018 injuries attended RIs for

**Table 1. Frequencies of motorcyclist road incidents-deaths and injuries-according to characteristics of the person, time and circumstance by sex.**

| Factor | Men | | | | Women | | | | Total | | | |
|---|---|---|---|---|---|---|---|---|---|---|---|---|
| | Deaths[a] | % | Injured | % | Deaths[a] | % | Injured | % | Deaths[a] | % | Injured | % |
| **n** | 363 | 84.8 | 64,849 | 73.7 | 65 | 15.2 | 23,122 | 26.3 | 428 | 100% | 87,971 | 100% |
| **Age group** | | | | | | | | | | | | |
| 1–14 | 5 | 1.4 | 918 | 1.4 | 2 | 3.1 | 644 | 2.8 | 7 | 1.7 | 1,563 | 1.8 |
| 15–29 | 205 | 56.5 | 38,577 | 59.5 | 28 | 43.1 | 13,508 | 58.4 | 234 | 54.6 | 52,085 | 59.2 |
| 30–49 | 135 | 37.2 | 22,365 | 34.5 | 31 | 47.7 | 8,117 | 35.1 | 165 | 38.6 | 30,483 | 34.7 |
| 50–64 | 13 | 3.6 | 2,687 | 4.1 | 4 | 6.2 | 759 | 3.3 | 17 | 4 | 3,447 | 3.9 |
| 65 and over | 5 | 1.4 | 301 | 0.5 | 0 | 0.0 | 93 | 0.4 | 5 | 1.2 | 395 | 0.4 |
| **Marital Status** | | | | | | | | | | | | |
| Unmarried | 197 | 54.3 | – | - - | 33 | 50.8 | - - | - - | 230 | 53.7 | - - | - - |
| Married / Consensual union | 94 | 25.9 | - - | - - | 19 | 29.2 | - - | - - | 113 | 26.4 | - - | - - |
| Widower/ Separated/ Divorced) | 2 | 0.6 | - - | - - | 1 | 1.5 | - - | - - | 3 | 0.7 | - - | - - |
| No data | 70 | 19.3 | - - | - - | 12 | 18.5 | - - | - - | 82 | 19.2 | - - | - - |
| **Condition** | | | | | | | | | | | | |
| Driver | 312 | 86.0 | 58,499 | 90.2 | 24 | 36.9 | 11,446 | 49.5 | 336 | 78.5 | 69,945 | 79.5 |
| Passenger | 29 | 8.0 | 6,350 | 9.8 | 35 | 53.8 | 11,676 | 50.5 | 64 | 15.0 | 18,026 | 20.5 |
| No data | 22 | 6.1 | 0 | 0.0 | 6 | 9.2 | 0 | 0.0 | 28 | 6.5 | 0 | 0 |
| **Incident class** | | | | | | | | | | | | |
| Crash | 286 | 78.8 | 30,199 | 46.6 | 57 | 87.7 | 10,017 | 43.3 | 343 | 80.1 | 40,216 | 45.7 |
| Occupant fall | 64 | 17.6 | 12,481 | 19.2 | 7 | 10.8 | 5,649 | 24.4 | 71 | 16.6 | 18,130 | 20.6 |
| Overturning | 3 | 0.8 | 3,216 | 5.0 | 0 | 0.0 | 1,322 | 5.7 | 3 | 0.7 | 4,538 | 5.2 |
| Run over | 8 | 2.2 | 3,190 | 4.9 | 1 | 1.5 | 638 | 2.8 | 9 | 2.1 | 3,828 | 4.4 |
| Fire | 0 | 0.0 | 9 | 0.0 | 0 | 0.0 | 1 | 0.0 | 0 | 0 | 10 | 0 |
| Other | 2 | 0.6 | 15,753 | 24.3 | 0 | 0.0 | 5,496 | 23.8 | 2 | 0.5 | 21,249 | 24.2 |
| **Year** | **Men** | | | | **Women** | | | | **Total** | | | |
| | Deaths | Rate[b] | Injured | Rate[b] | Deaths | Rate[b] | Injured | Rate[b] | Deaths | Rate[b] | Injured | Rate[b] |
| 2012 | 92 | 8.2 | 14,948 | 1,326.8 | 14 | 1.1 | 4,739 | 374.2 | 106 | 4.4 | 19,687 | 822.7 |
| 2013 | 102 | 9.0 | 16,455 | 1,446.0 | 12 | 1.0 | 6,036 | 471.8 | 114 | 4.7 | 22,491 | 930.4 |
| 2014 | 90 | 7.8 | 16,017 | 1,394.0 | 20 | 1.6 | 5,636 | 436.2 | 110 | 4.5 | 21,653 | 887.0 |
| 2015 | 79 | 6.8 | 17,429 | 1,502.8 | 18 | 1.4 | 6,711 | 514.4 | 97 | 3.9 | 24,140 | 979.6 |
| **Period Average (95% UI)[c]** | 91 | 7.9 | 16,212 | 1,417.4 | 16 | 1.3 | 5,781 | 449.2 | 107 | 4.4 | 21,993 | 904.9 |
| | (74–97) | (5,9–8,6) | (13,313–17,602) | (1,181–1,516) | (14–19) | (1.2–1.5) | (4,100–6,805) | (321–522) | (91–113) | (3.7–4.6) | (17,417–24,404) | (726–990) |

**Sources:** database of the RUAF-D and IPAT ("*in situ*").

[a] cases adjusted for underreporting;—-absence of information

[b] per hundred thousand inhabitants

[c] Period 2012–2015, Uncertainty Intervals calculated with the Bootstrap technique in 10,000 samples.

motorcyclist; 68.8% of the events occurred in men, with a higher concentration in the age groups of 15–29 years (52.4%, n: 23,603) and 30–49 years (36.3%, n: 16,347). When asked about the type of affiliation to the social security system in Colombia, it was found that 25.5% were affiliated to the contributory system (n: 11,449). 88.6% (n: 39,902) of the events were attended in the outpatient service, 10% (n: 4,496) in the emergency department and 1.4% (n: 620) in-hospital. 93.3% (n: 41,999) of the cases had only one nature of injury of interest associated with them and 6.7% (n: 3,019) had more than two injuries; contusions, other muscle and tendon injuries, open wounds and severe traumatic brain injuries were the higher frequent.

## Years of Life Lost due to premature death (YLLs) due to RIs of motorcyclists

Of the total burden of disease due to RIs of motorcyclists, premature death constituted 33.4% in the four-year period (26,705 YLLs), with an average of 6,676 (95% UI: 5,682–7,032) per year, which corresponded to a rate for the period of 274.8 YLLs per 100,000 inhabitants. Throughout the period, both men and women had a stable loss with a downward trend, this being greater for women in 2013 (reduction of 18%) and for men in 2015 (reduction of 13.6%), compared to the previous year (S3 Table).

Of the YLLs due to RIs, 94.5% (25,251) were concentrated in young people –15 to 29 years old– (60.7%), and young adults –30 to 49 years old– (33.8%), with ratios of approximately 7:1 and 5:1 compared to women in the abovementioned groups, respectively, with a marked decrease in YLLs rates from the age of 29, for both men and women, a behavior that was similar in each of the years studied (Table 2).

## Years Lived with Disability (YLD) due to RIs of motorcyclists

Of the total burden of disease due to RIs of motorcyclists in the city of Medellin, morbidity constituted 66.6% in the four-year period (53,342 YLDs), with an average of 13,335, (95% UI:

**Table 2. YLLs, YLDs and DALYs due to road incidents of motorcyclists according to sex and age group.**

| Sex / age group | YLLs | | | YLDs | | | DALYs | | |
|---|---|---|---|---|---|---|---|---|---|
| | n | % | Rate [a] | n | % | Rate [a] | n | % | Rate [a] |
| **Men** | | | | | | | | | |
| 1–14 | 423 | 1.9 | 45.6 | 6 | 0.0 | 0.6 | 429 | 0.7 | 46.2 |
| 15–29 | 14,196 | 62.5 | 1212.3 | 26,740 | 63.1 | 2,283.5 | 40,936 | 62.9 | 3,495.8 |
| 30–49 | 7,363 | 32.4 | 578.9 | 13,499 | 31.8 | 1061.3 | 20,862 | 32.0 | 1,640.2 |
| 50–64 | 517 | 2.3 | 62.7 | 1,443 | 3.4 | 175.1 | 1,960 | 3.0 | 237.8 |
| 65 and older | 201 | 0.9 | 53.2 | 721 | 1.7 | 190.3 | 922 | 1.4 | 243.5 |
| Subtotal | 22,701 | 34.9 | 496.3 | 42,409 | 65.1 | 927.2 | 65,110 | 81.3 | 1,423.5 |
| (95% UI Sub)[b] | (19,980–24,105) | | | (37,740–44,567) | | | (62,210–67,320) | | |
| **Women** | | | | | | | | | |
| 1–14 | 159 | 4.0 | 17.9 | 5 | 0.0 | 0.6 | 164 | 1.1 | 18.5 |
| 15–29 | 2,025 | 50.6 | 175.6 | 6,634 | 60.7 | 575.2 | 8,659 | 58.0 | 750.8 |
| 30–49 | 1,666 | 41.6 | 111.7 | 3,482 | 31.8 | 233.4 | 5,148 | 34.5 | 345.0 |
| 50–64 | 154 | 3.8 | 14.7 | 492 | 4.5 | 47.1 | 646 | 4.3 | 61.8 |
| 65 and older | 0 | 0.0 | 0.0 | 320 | 2.9 | 56.6 | 320 | 2.1 | 56.6 |
| Subtotal | 4,004 | 26.8 | 77.8 | 10,933 | 73.2 | 212.6 | 14,937 | 18.7 | 290.4 |
| (95% UI Sub)[b] | (3,504–4,689) | | | (8,659–12,344) | | | (12,245–16,678) | | |
| **Both genders** | | | | | | | | | |
| 1–14 | 582 | 2.2 | 32.1 | 11 | 0.0 | 0.6 | 593 | 0.7 | 32.7 |
| 15–29 | 16,222 | 60.7 | 697.9 | 33,373 | 62.6 | 1,435.8 | 49,595 | 62.0 | 2,133.8 |
| 30–49 | 9,029 | 33.8 | 326.7 | 16,981 | 31.8 | 614.4 | 26,010 | 32.5 | 941.1 |
| 50–64 | 671 | 2.5 | 35.9 | 1,935 | 3.6 | 103.5 | 2,606 | 3.3 | 139.4 |
| 65 and older | 201 | 0.8 | 21.3 | 1,041 | 2.0 | 110.2 | 1,242 | 1.6 | 131.5 |
| Total | 26,705 | 33.4 | 274.8 | 53,342 | 66.6 | 549.0 | 80,046 | 100.0 | 823.8 |
| (95% UI Tot)[b] | (22,119–27,890) | | | (50,178–55,676) | | | (77,502–82,160) | | |

[a] per 100,000 inhabitants.

[b] Period 2012–2015, Uncertainty Intervals calculated with the Bootstrap technique in 10,000 samples.

8,302–16,905) per year, which corresponded to a rate of 549.0 YLDs per 100,000 inhabitants. In general, the proportion of YLDs was higher in men, 79.5% and in each of the age groups, with marked differences of almost 38 percentage points in young people from 15 to 29 years old, and of 19 percentage points in young adults, from 30 to 49 years old; the two previous age groups contributed 94.4% (50,354) of the YLDs due to RIs of motorcyclists; after 49 years of age, a progressive decrease in the rate of YLLs was observed (Table 2). The previous results had the same pattern per year; particularly in 2015, YLDs increased 113% compared to 2012 and 11% compared to 2014.

According to the number of injuries associated with each event, the burden of morbidity (YLDs) was higher for those motorcyclists who suffered two bodily injuries (65.0%)—standing out among them, severe traumatic brain injuries and a contusion in any part of the body. Then, in order of magnitude, there were those who had three injuries (28.9% of YLDs) related to severe traumatic brain injuries, contusions in any part of the body, and open-wound injuries. For those who suffered a single injury (3.4% of YLDs), this was related to severe traumatic brain injuries. Of the YLDs for the RIs of motorcyclists in which four specific injuries derived from the outcome 2.7% were, in addition to those indicated above, related to upper and/or lower limb fractures (S4 Table).

## Disability-Adjusted Life Years (DALYs) due to RIs of motorcyclists

In the period 2012–2015, RIs of motorcyclists caused, in absolute terms, 80,046 DALYs (rate of 823.8 DALYs per 100,000 inhabitants) (Table 2). The most noticeable change was observed in 2014 compared to the previous year, where the increase was 24.5% in the DALYs rate and 25.6% in absolute terms (S3 Table).

Due to RIs of motorcyclists, the burden mainly affected men (81.3%) at a ratio of 1:5 and at a rate of 1,423.5 DALYs per 100,000 men. Young motorcyclists and young adults were disproportionately affected by RIs, at 94.5% (n: 75,605) of loss of DALYs with a higher concentration in victims aged 15 to 29 years. When considering DALYs rates by age groups and sex, men had a more significant contribution to the loss than women in the 15–29 and 30–49 age groups. In both sexes, a constant decrease in DALYs was observed from the age of 29; however, in men over 65 years of age, an increase of 3% was identified compared to those aged 50–64 years (Table 2).

## Discussion

Injuries resulting from RIs of motorcyclists were mistakenly thought to be a problem primarily affecting industrialized countries; however, statistics have shown that many "diseases of the developing world" are occurring in low-income countries, adding to the burden of poverty-related diseases [21]. The urban environment in countries with social inequalities stimulates the growth of the automotive industry and the use of motorcycles in particular as a means of transport and work [22]. This trend has been observed in several countries, including Colombia, where the situation has worsened due to the ease and low costs at which this means of transport is acquired [22, 23]. In Medellin, the burden of morbidity and mortality generated by motorcycle RIs has become a public health problem given its high frequency and the traumatic effects it triggers on the population. The conveniences derived from the use of the motorcycle are recognized in terms of the reduction in monthly fixed expenses for many households in the city, in travel times, among others [23]; but the negative impacts on other aspects related to pollution of the environment and probability of fatal and non-fatal road accidents [24] are also evident.

The road safety policy has traditionally been aimed primarily at reducing the number of fatal victims; however, RIs also cause a large number of non-fatal injuries, generating considerable economic and human costs. In Medellin, while the number of motorcyclists who have died from RIs suggests moderate behavior, the cases of injured are unstable with an upward trend. This situation is in contrast to that reported by other studies [25, 26], which suggest a slower decline in the number of injured than in fatalities, and even an increase in other countries.

During the four years of study, the loss of healthy life years due to motorcyclist road incidents in Medellin was 80,046 DALYs (in absolute terms), which represented a rate of 823.8 DALYs per 100,000 inhabitants. When compared to Colombia's rate for 2019 according to WHO, 319 per 100,000, the annual rate of disability-adjusted life years (DALYs) is only 2.6 times higher than the WHO estimate. To consider these differences, it should be kept in mind that the studies conducted worldwide on burden disease are based on predictive models that include meta-analysis and Bayesian models.

Between 2012 and 2015 in Medellin, most of the loss of healthy years of life, DALYs, per RIs of motorcyclists was due to disability (YLDs; 66.6%; 549 per 100,000 population) with a higher concentration in men at a 4 to 1 ratio compared to women. However, other similar studies, albeit with estimates of the indicators with different metrics from those of this study, showed affinity with the findings of this research. In Thailand, for example, according to reports from 2004, 80% of accidents involved motorcycles; and of these, it was learned that the DALYs rate due to RIs was 1,080 per 100,000 persons per year (1,780 and 390 for men and women, respectively), and 2,100 per 100,000 among those aged between 15 and 29 years [27]. In Nepal, the estimated burden for 2014–2015 was 567 DALYs per 100,000 inhabitants, with a predominance of men, at 73%, especially with ages 15–29 and 30–49 years [28]. In Europe, the Netherlands estimated a rate of 470 DALYs per 100,000 inhabitants, of which 55% were motorcyclists [25].

A study in 2015 reported that, in the southern cone, Colombia along with Argentina, Chile and Peru were the countries that had the lowest rates of DALYs due to RIs in general, between 615 and 700 per 100,000 inhabitants, in contrast to Paraguay and Brazil, which were the most affected (1,270 and 1,230 DALYs per 100,000 inhabitants, respectively). However, according to IHME's reports in 2019, when focusing on the comparison of DALYs due to RIs of motorcyclists, Colombia ranked third with a rate of 319 DALYs per 100,000 persons, after Paraguay and Brazil [29, 30]. However, these results do not allow comparability given the difference between the methodologies used, as mentioned before.

In most countries that do not have a culture of information, problems are accessing reliable sources of information with good coverage, which generates errors in mathematical modelling and, therefore, derived estimates [31]. In the case of Medellin, however, it was possible to estimate the epidemiological parameter by nature of injury associated with motorcyclist road incidents through secondary information sources generated from vital statistics and health care records; therefore, it was not necessary to develop prediction processes, which would provide greater reliability to the results obtained. Because about 7% of the events attended in the health services presented more than two natures of injury generated by the same road incident, the calculation of the DALYs was adjusted through a multiplicative model [15], which allowed this indicator to reflect the sum of the total health lost in the individual. There was no double or even quadruple counting of this loss.

Contrary to what was found in other studies on the burden of injuries due to RIs [27, 28, 32], it was observed that the greatest loss of years of healthy life in motorcyclists in Medellin was due to non-fatal injuries. This was reported in other studies focused on identifying high-risk groups. The studies also disaggregated their analyses into different age groups and means

of transport [25, 33]. In most age groups, YLDs had the greatest contribution to DALYs, standing out among young people as well as motorcyclists (55% of the burden resulted from non-fatal injuries).

These differences could be explained, firstly, by the fact that most of the reviewed studies declared an underestimation in the number of YLDs; secondly, due to the methodological differences and severity levels included in the studies since several of them only focused on injuries of 2 or more in the maximum score of abbreviated injuries and taken care for only in hospitalization services, which leads to an underestimation of the impact of injuries on the population [25]; and third, because it has been observed that the distribution of RIs by victim's means of transportation is substantially different for fatalities and non-fatal cases [34], where the number of motorcycle drivers in inpatient admissions and outpatient visits, when compared to other victims, is vastly higher [25, 34–36]. In consistency with the foregoing, it has been documented that the burden of injuries for different means of transport may vary between and within countries, probably due to differences in injuries found and differences in age distribution. Therefore, it should not be surprising that other studies report different results [32].

The concentration of fatal and non-fatal injuries (YLLs and YLDs) in the age group from 15 to 49 years old was striking, with an important contribution found in young people (15 to 29 years old) probably because at these ages, to a greater extent, irresponsible behaviors are assumed on the road that can cause disability or even death, with the aggravating factor that in the first case survival from the injuries does not guarantee absence of sequelae, which if it occurs, will negatively impact the quality of life [36]. In this regard, other factors that have been related to the higher death rates due to RIs, in general, in drivers under 25 years of age, are the lack of experience, unsafe attitudes, sensation seeking, lower risk perception and lower use of safety measures [37].

## Conclusions

It is concluded that the greatest contribution to the loss of years of healthy life due to RIs of motorcycles (DALYs) was due to non-fatal injuries, which were concentrated in young men, which emphasizes the impact on the economically active and reproductive population.

These results suggest that identifying high-risk groups based solely on road fatalities may overlook groups that are at risk. While fatal incidents receive a lot of attention and are often studied in detail, incidents that only result in injuries are hardly documented and studied. Consequently, little information is available on their magnitude, distribution, and causal factors, which are essential for the development of effective prevention strategies.

It is recommended to work on a road safety plan for motorcyclists in order to achieve a common vision that enables authorities, considering the multiplicity of agents and sectors of action, to agree on objectives and on appropriate and agreed action commitments. It is necessary for Road Safety Plans to be incorporated into the strategic lines of action, actions to improve response times, the development of protocols for specialized care for motorcyclists, and the coordination of emergency services for the rescue and transfer of the wounded to hospital institutions.

The quality of information and the articulation of the information systems involved in the care, monitoring, and control of these events should be a topic to be studied in depth so that it allows decision makers to direct resources and reduce the lag compared to the objectives of the Global Plan for the Decade of Action for Road Safety [38]. The importance of a single, systematized, and accessible epidemiological surveillance on RIs is emphasized.

For the benefit and accuracy of future studies, it is convenient to improve the process used to identify the events of injuries taken care for, for which several of the criteria assessed by Pérez *et al.* may be considered [39]. It is essential to work on the estimation of disability weights that make it possible to assess the local magnitude of the problem and, in the absence of an injury classification system incorporated in the health services, begin the exploration of programs that allow the derivation of severity scores from the ICD-10 codes [25, 39].

## Limitations and further studies

The limitations of this study that stand out would have to be those related to the secondary sources of information used. Regarding mortality, vital statistics can pose coverage problems due to the eventual incomplete registration or misclassification of deaths. The research was taken from several sources of information, such as RUAF-D, SSSA, Secretary of Mobility, Secretary of Health of Medellin, IPAT and RIPS. These entities respond to the monitoring of each event independently rather than working together to connect information cohesively. This prevents optimal traceability and presents differences that could show gaps concerning the registry, which consequently makes the reliability and representativeness of the information available in the official mortality databases questionable. However, with the adjustment for under-registration, which is quite low, this impasse was corrected, understanding the contents of the statement of Colombian National Administrative Department of Statistics (DANE, for the Spanish acronym), in the sense that under-registration in mortality and the omission in certification for these causes is much lower than that of natural causes, because several instances intervene in their investigation, and the vital statistics system captures a number of those deaths that the competent authorities do not know.

As neither of the two sources of information were available, both possessing a unique identifier necessary to apply some of the available methods for morbidity (e.g., the capture-recapture method [40]), the events related to the injured could not be adjusted due to underreporting.

Similarly, the source of information used to identify the injured treated in the health service provider institutions (IPS, by its Spanish acronym) were the RIPS, which present limitations in terms of the quality of the diagnostic codes they record. Although those injuries are of mandatory notification, many IPS may not make the report or don't make it in a timely fashion, which generates that the consolidated ones are updated to one or more years. Notwithstanding, this data source was chosen because it is the one that provides the most structured data about the events that lead to consultation in an IPS at the national level, specifically the RIPS related to traffic accidents; given its legal and economic implications, the information provided by the Compulsory Traffic Accident Insurance (SOAT, by its Spanish acronym) has greater integrity of the data reported, and according to studies conducted by the National Institute of Health, it is possible to use it for purposes related to surveillance in public health.

As stated by the National Institute Legal of Medicine and Forensic Sciences [41], in the morbidity analysis, injuries expressed in terms of rates may not represent an approximation to the complete phenomenon of road insecurity because, among other reasons, the report of injured persons due to transport accidents is a process of ongoing discussion at the global level given the complexity of the assessment and the definition of the concept of injured or wounded. The different forms of counting are a limiting factor for a rigorous, correct, and comparable analysis of non-fatal victims of road accidents, taking into account that the scope of the forensic service of Legal Medicine does not cover all the injured persons by traffic accidents of the country and only serves the purposes of the justice system. It, therefore, reaches

only part of the population that is estimated at about 10% according to the proportion of injured per death in the WHO world projections.

The use of a composite indicator of health status such as the DALYs is based on a series of assumptions and estimates; thus, the paucity of local data and variations in study types and standards make calculating an accurate estimate of burden due to injuries extremely difficult. In the absence of a representative data set at the city level, the source of information that provided greatest coverage and information about non-fatal events had to be used with a methodological adjustment that allowed for the identification of motorcyclists, among whom the means of transport was unknown. For this reason, the results of YLDs may be underestimated at the population level since the cases of injuries by nature of injury were obtained from patients with access to health services and some with minor injuries may not have visited any IPS. Similarly, the estimates were made based on each year of study as an approximation to the prevalence by nature of injury, which could reflect the burden in each period while underestimating the real burden.

Another limitation identified is a possible duplication for those subjects who died after their hospital stays, as it was not possible to identify them in the anonymized database of deaths; thus, the results may be presenting an overestimation, perhaps 5%, as has been indicated in other studies [34].

## Practical implications

This research was carried out to establish the magnitude of the impact of road safety on motorcycle users by calculating the years of healthy life lost. The motorcycle accounts for 57.1% of the automotive fleet in Colombia, and has become a fundamental aspect for economic dynamics as it is a tool for the employment of approximately 2.6 million people. Between 2010 and 2018, the motorcycle market grew by 47%, with an average annual rate of approximately 8%. This phenomenon means more population at risk and a potential increased burden of disease manifested in trauma, injury and death.

According to our results, it is pertinent to adopt and / or adapt some of the successful educational measures such as the generation of gradual access systems to driving according to age, experience of the driver, size and power of the motorcycle.

An important challenge is to mobilize the population in pursuit of safe attitudes and solidarity in compliance with the law and responsibility in traffic, reinforcing the idea that RIs should not be considered inevitable but be prevented and controlled through effective measures to prevent citizens from becoming victims when they exercise their fundamental right to come and go. Despite the existence of legislation in Colombia related to the mandatory use of helmets as elements of road safety for the population, problems have been encountered in its compliance, first for climatic, aesthetic and hygienic reasons, and second, because of the perception of risk and the possibility of being stopped and sanctioned by the authorities is increasingly lower. Among the few studies [42, 43] that have been carried out to evaluate the effectiveness of road safety measures, it has been identified that the helmet is not internalized in the population as an object that provides safety and protects life. It is used to avoid a traffic ticket, being used intermittently and inappropriately. Moreover, its use is differential by geographic regions, where social and economic development is key factors. Thus, the call is for everyone to commit themselves to take care of their own lives and the lives of others through their actions and responsibilities.

The combination of three factors, human, environmental, and automotive, constitute the fundamental elements to be considered in road events. In particular, knowledge of the human factor is essential in road safety to guide intervention strategies and prevent accidents and

their causes. The human factor is the most difficult to control and the most influential because driving skills and habits are decisive for accident prevention. Although in theory, drivers, in general, should be in optimal psychophysical conditions when driving, in practice, this is not usual, and therefore deficiencies in psychophysical health can trigger events that can life-threatening to drivers, passengers, and pedestrians. In particular, studies addressing the human factor analysis in road incidents involving motorcyclists are scarce, probably due to the complexity of the evaluation of human characteristics.

Some studies have focused on the human factor in motorcycle drivers involved in RIs in Latin American countries. A descriptive study in Paraguay for this specific population concluded that the greatest attribution to the occurrence of road events was due to behavioral factors [44]. In Argentina, the costs and factors derived from motorcycle accidents were analyzed, highlighting that the human factor was the leading cause, especially the lack of respect for traffic regulations, within which speeding when performing maneuvers can cause harm to the driver and passengers [45]. In Caracas, in a case-control design, a positive association was found between the presence of Attention Deficit Hyperactivity Disorder (ADHD) and a higher occurrence RIs of motorcyclist in people between 18 and 63 years of age, i.e., motorcycle drivers who presented the disease had a higher risk of experiencing RIs incidents due to their greater inclination to violate traffic laws and to have inappropriate behavior, in comparison to motorcycle drivers who did not present ADHD [46].

In Medellin in 2015 [47], the psychological profile of drivers who repeatedly violated traffic rules were analyzed, identifying that reckless, risky, and aggressive driving, disrespect for traffic rules, emotional instability, especially anxiety in young drivers, and dissociative personality were related to attentional errors in driving and traffic incidents. The greater impact on road events in the young population could be explained by their perception of risk, which is also affected by perceptual abilities, driving experience, emotional state, and traffic regulations [48].

In particular, the trait of road rage, as a human factor, has been little studied in the context of professional drivers. Useche et al., in 2019 [49], analyzed the association between trait road rage and driving styles in a sample of Colombian professional drivers, highlighting, among the main findings that impeding forward progress by other drivers, illegal driving and hostility significantly predicted adaptive and maladaptive driving styles. It was concluded that interventions that focus on minimizing the driving anger profile of drivers could be effective in counteracting the consequences of such predisposition. Apart from the behavioral problems referred to above, motorcycle drivers in the city of Medellin are known for their special predisposition to contravene road regulations. In a comparative study on the profile of motorcyclists in four Colombian cities, Bogotá, Medellin, Cucuta, and Bucaramanga [50], it was found that it is in Medellin where this type of driver violates the National Traffic Code to a greater extent, and it was concluded that the motorcyclist's profile is different depending on the region they are from and therefore it is ratified that generalized public policies cannot be carried out by local governments towards issues related to motorcycles.

The human factor is the most difficult to control and is influential since driving skills and habits determine accident prevention. In this sense, behavioral interventions are essential for structuring safe road systems and, therefore, require the design of intervention programs to change driver behavior [51]. Necessarily, in the perspective of minimizing the risks inherent to the human factor, prior knowledge of the causal factors is required through antecedent and consequent communication strategies with the driver and the application of principles of social influence, which allow creating suitable conditions for the desired behavior to occur.

This study is a pioneering effort in Colombia because it refers to the experience of the burden of disease caused by motorcycle RIs with the application of the new methodology

proposed; despite the merit of this approach, the lack of studies in the setting where the new metric is applied becomes a disadvantage given the impossibility of carrying out a comparative analysis, even though for the outcome of interest in this study, the RIs, the indicators have similarities.

## Supporting information

**S1 Table. Nature of lesions and assigned ICD10 codes.**
(DOCX)

**S2 Table. Disability weights by nature of injury according to GBD.**
(DOCX)

**S3 Table. Deaths, injuries in situ, injuries attended, YLL, YLD and DALY per year, according to sex and age groups.**
(DOCX)

**S4 Table. YLD according to the nature of the injury and the number of injuries per event.**
(DOCX)

**S1 Dataset. Database of the unique registry of affiliates-module of deaths (RUAF-D, for the Spanish acronym).**
(XLSX)

**S2 Dataset. Database of injured persons according to police traffic accident reports (IPAT, for the Spanish acronym).**
(XLSX)

**S3 Dataset. Database of individual registries of the provision of health services (RIPS, for the Spanish acronym).**
(XLSX)

## Acknowledgments

To the Health Information System Management professional, Dorian Ospina Galeano, for her support in reviewing the morbidity and mortality databases, especially in their quality control.

## Author Contributions

**Conceptualization:** Sandra Milena Porras Cataño, Hugo Grisales-Romero.

**Data curation:** Sandra Milena Porras Cataño, Hugo Grisales-Romero.

**Formal analysis:** Sandra Milena Porras Cataño, Hugo Grisales-Romero.

**Funding acquisition:** Sandra Milena Porras Cataño, Hugo Grisales-Romero.

**Investigation:** Sandra Milena Porras Cataño, Hugo Grisales-Romero.

**Methodology:** Sandra Milena Porras Cataño, Hugo Grisales-Romero.

**Project administration:** Sandra Milena Porras Cataño, Hugo Grisales-Romero.

**Resources:** Sandra Milena Porras Cataño, Hugo Grisales-Romero.

**Software:** Sandra Milena Porras Cataño, Hugo Grisales-Romero.

**Supervision:** Sandra Milena Porras Cataño, Hugo Grisales-Romero.

**Validation:** Sandra Milena Porras Cataño, Hugo Grisales-Romero.

**Visualization:** Sandra Milena Porras Cataño, Hugo Grisales-Romero.

**Writing – original draft:** Sandra Milena Porras Cataño, Hugo Grisales-Romero.

**Writing – review & editing:** Sandra Milena Porras Cataño, Hugo Grisales-Romero.

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
