## [Decision Letter · Decision Letter 0]

16 Apr 2021

PONE-D-20-40988

Loss of Years of Healthy Life Due to Road Injuries of Motorcyclists in the City of Medellín, 2012 to 2015

PLOS ONE

Dear Dr. Grisales-Romero,

Thank you for submitting your manuscript to PLOS ONE. After careful consideration, we feel that it has merit but does not fully meet PLOS ONE’s publication criteria as it currently stands. Therefore, we invite you to submit a revised version of the manuscript that addresses the points raised during the review process.

First of all, apologies for the delay in the editorial decision made in regard to the first version of your paper. Linked to the current situation, our referees needed some additional time to complete their reports.

Your paper has been scientifically judged by two acknowledged experts in this field, that (although finding merit and potential in the study) have raised an extensive set of comments and suggestions, whose resolution is needed for considering the acceptance of the paper in PLOS ONE. Although most of the comments provided by our reviewers are considerably amendable (e.g., explicitly stating the study aim, improving the presentation of results and their discussion), some other issues require major attention. Please refer to all the comments appended below for your revisions.

We look forward to receiving your revised manuscript.

Kind regards,

Sergio A. Useche, Ph.D.

Academic Editor

PLOS ONE

Journal Requirements:

We note that you have included text in the title page that is not in English, as per the journal guidelines (https://journals.plos.org/plosone/s/submission-guidelines) manuscripts must be submitted in English, please therefore update the title page text to English.

We note you have included a table to which you do not refer in the text of your manuscript. Please ensure that you refer to Table 1 in your text; if accepted, production will need this reference to link the reader to the Table.

Please include captions for your Supporting Information files at the end of your manuscript, and update any in-text citations to match accordingly. Please see our Supporting Information guidelines for more information: http://journals.plos.org/plosone/s/supporting-information.

Reviewers' comments:

Reviewer's Responses to Questions

**Comments to the Author**

1. Is the manuscript technically sound, and do the data support the conclusions?

Reviewer #1: Partly

Reviewer #2: Partly

2. Has the statistical analysis been performed appropriately and rigorously? 

Reviewer #1: Yes

Reviewer #2: I Don't Know

3. Have the authors made all data underlying the findings in their manuscript fully available?

Reviewer #1: Yes

Reviewer #2: No

4. Is the manuscript presented in an intelligible fashion and written in standard English?

Reviewer #1: Yes

Reviewer #2: Yes

5. Review Comments to the Author

Reviewer #1: The authors perform a simple, descriptive study on the burden of road injures that occurred in Medellin between the years 2012 and 2015. First, I would like to acknowledge the authors for the arduous work performed in the curation and analyses of such a big amount of data. Although the design is completely valid and the information presented may be of interest to the Colombian authorities, many issues should be addressed before considering the paper for publication in PLoS ONE.

General comments:

1. I am missing explicit information on the exact purpose of the study.

2. Please, include as much information as possible about the population/sample of the study, cases analyzed, sociodemographics. It would be of great help to have the materials and methods sections divided into subsections.

3. I recommend the authors include a “limitations and further studies” (or similar) section in which the potential analyses not performed in this study (inferential comparisons, prediction) and other matters could be addressed.

4. Also, I would find it interesting and useful as a reader to have a “practical implications” section.

5. few format issues should be addressed (e.g., some paragraphs without indentation as in line 236, line spacing in paragraph 176). One could more comfortably read the article with a homogeneous format. Please, review the general format of the paper including the tables.

Specific comments:

6. I encourage the authors to rewrite the first sentence (lines 46-47). If the authors want to highlight this terminological clarification as in the reference provided, I believe further explanation could be given to ease the readers with the understanding of the matter of study. Maybe there is a conceptualization misunderstanding between road injuries and road crashes (see https://doi.org/10.18270/cuaderlam.v16i30.2842). A car may be involved in a traffic crash but injured is not a term usable for cars, but persons instead. The use given to the term road injuries (of Motorcyclists) in the title would be appropriate. Please, review it throughout the paper.

7. Line 52, I would give this source of information as a reference (instead of as text) to make it easier for the reader to access the data.

8. Line 65, GDP, when abbreviations are used for the first time the full meaning of the abbreviation should be given. Also, ICD-10.

9. There are several lines (e.g., 221, 241, 301) where “Error! Reference source not found” is written, correct this.

10. The references section should be thorough reviewed. There are a considerable number of format and citing errors that must be addressed. Please follow the referencing style guidelines of the journal.

10.1 Please, do not use both the Spanish and English acronym for World Health Organization.

10.2 Please, translate Spanish terms into English where appropriate as required by normative.

10.3 Do not cite journal articles as webpages.

10.4 Do not include the term “volume” before the number of the volume.

10.5 The month and day of publications are not required.

Reviewer #2: This interesting study aims to estimate disease burden in terms of disability-adjusted life years (DALYs) lost due to RIs of motorcyclists in the city of 96 Medellín between 2012 and 2015. The information sources were the Single Register of Affiliates, death certificates from the vital record office (RUAF-D), the database of injured persons according to the Police reports of traffic accidents at the scene ‒“in situ”‒, provided by the Mobility Secretariat of Medellín and the Individual Registries of Provision of Services (RIPS) of the outpatient, hospitalization, and emergency services. Of the total number of RIPS of the outpatient, hospitalization, and emergency services, a selection was made which included the records containing (as a diagnosis—main or related) an ICD-10 code associated at least with one of the injuries considered [11]; The analysis of YLL (using average life expectancy values ) and YLD for single injuries followed standard global burden of disease estimates however the YLD adjustment for multiple injuries sustained by the same patient appears novel. According to death registrations and police reports there were 428 deaths over the study period, 80% in men and 87,971 injury incidents.

The DALYs analysis, a rate of 80,000 DALYS per 100,000 over the period of study is not directly comparable to the WHO estimate of 319 per 100,000 for Motorcycle incidents in Colombia in 2019. However as the latter has a lifetime horizon for YLD it would be expected to be higher. This does call into question the methods used in this study as the DALYS annual rate is per year 60 times greater than the WHO estimate, and also suggests that most of the disease burden came from non fatal injuries which runs contrary to GBD studies.

The findings are interesting however the following clarifications are needed.

Data on about 3,000 patients has been made available in the submission but there is no statement on where the other data sources for mortality and incidents can be accessed

Methods:

Please clarify if and how the police reports were linked to hospital data to ensure accurate estimates of non fatal injury and address the data availability queries above

Results:

Produce a further descriptive table of the 87,971 incidents detailing how many resulted in hospital admission, non presentation to healthcare, or outpatient attendance only cross tabulating frequencies of injuries sustained in those killed or admitted – TBI, spinal injury, thoraco abdominal injury, limb injury.

Discussion:

Defend the use of the novel calculation of YLD for multiply injured patients, comment further on the large discrepancy between these estimates and those of WHO in 2019.

Comment on the use of helmets by motorcyclists in Colombia for context

6. PLOS authors have the option to publish the peer review history of their article (what does this mean?). If published, this will include your full peer review and any attached files.

Reviewer #1: No

Reviewer #2: **Yes: **Fiona Lecky

---

## [Author Response · Author response to Decision Letter 0]

8 Jun 2021

The response to the reviewers is in file: Response to Reviewers.docx

---

## [Decision Letter · Decision Letter 1]

2 Jul 2021

PONE-D-20-40988R1

Loss of years of healthy life due to road incidents of motorcyclists in the city of Medellin, 2012 to 2015

PLOS ONE

Dear Dr. Grisales-Romero,

Thank you for submitting your manuscript to PLOS ONE. After careful consideration, we feel that it has merit but does not fully meet PLOS ONE’s publication criteria as it currently stands. Therefore, we invite you to submit a revised version of the manuscript that addresses the points raised during the review process.

Your manuscript has been reviewed for a second time. Overall, the reviewer endorses most of the changes made, asking you to perform some further amendments (most of them slight) in the manuscript. Personally, and after a careful read of your revised paper, I would also like to ask you to strengthen the literature review, specifically in regards to the empirical evidences on human factors and road safety, e.g., risky road behaviors in the context of Medellín, or at least Colombian/Latin American cities, and the relationship between the specific road environment of the country and toad behaviors and crashes. This is especially problematic if the fact that most of the papers empirically documenting this issue were not considered in your manuscript, and many of the study assumptions were done on the basis of non-scientific sources. In other words, in its present form, the paper lacks from a clear scientific support on the road safety panorama of Colombian drivers (including motorcyclists).

In order to expedite the processing time of your manuscript, I will take an editorial decision after receiving your revised paper (without a new round of reviews), in case I find the improvements are adequate and complete.

We look forward to receiving your revised manuscript.

Kind regards,

Sergio A. Useche, Ph.D.

Academic Editor

PLOS ONE

Journal Requirements:

Additional Editor Comments (if provided):

Reviewers' comments:

Reviewer's Responses to Questions

**Comments to the Author**

1. If the authors have adequately addressed your comments raised in a previous round of review and you feel that this manuscript is now acceptable for publication, you may indicate that here to bypass the “Comments to the Author” section, enter your conflict of interest statement in the “Confidential to Editor” section, and submit your "Accept" recommendation.

Reviewer #1: All comments have been addressed

2. Is the manuscript technically sound, and do the data support the conclusions?

Reviewer #1: (No Response)

3. Has the statistical analysis been performed appropriately and rigorously? 

Reviewer #1: (No Response)

4. Have the authors made all data underlying the findings in their manuscript fully available?

Reviewer #1: (No Response)

5. Is the manuscript presented in an intelligible fashion and written in standard English?

Reviewer #1: (No Response)

6. Review Comments to the Author

Reviewer #1: All comments have been appropriately addressed. I am only suggesting a few minor corrections to improve the article.

Please, add a reference in the first paragraph.

Lines 46-48, consider rewriting the sentence from “In this phenomenon” to “combined.”.

Line 121-123, consider rewriting.

Line 257, consider adding the percentage for unmarried cases.

Line 261 and 290, add .0 to the 82% and 98%

Line 272, IQR should be only a number as it presents de distance between quartiles 25th and 75th. If the authors are expressing the quartiles 25 and 75, please correct it.

Line 291, I am not sure what this 25% inside the parenthesis stands for.

Line 437 and 440, Consider adding a full stop instead of a semicolon before “in consistency” and “therefore”, respectively.

References, I encourage the authors to carefully check all the references.

I would remove the PMID if already giving the doi.

Reference 36, Injury Prevention is the name of the journal I am not sure what does 6.4 stands for, please check it and correct it if necessary.

Reference 39, “Ciênc. saúde colet” should be each word capitalized as it is the name of the journal. Same for reference 28.

7. PLOS authors have the option to publish the peer review history of their article (what does this mean?). If published, this will include your full peer review and any attached files.

Reviewer #1: No

---

## [Author Response · Author response to Decision Letter 1]

11 Aug 2021

Reviewer #1's comments

Response: We have included the requested elements in this resubmission of the manuscript.

2. Please, add a reference in the first paragraph.

Response: We have included a reference in the first paragraph (Line 50)

3. Lines 46-48, consider rewriting the sentence from “In this phenomenon” to “combined.”.

Response: We corrected the text (Lines 46-47).

4. Line 121-123, consider rewriting.

Response: We rewrote the paragraph. This is the improved version: “These were all records of deaths and injuries due from RIs of motorcyclists, collected from different sources, in the city of Medellin between 2012 and 2015” (Lines 133-134).

5. Line 257, consider adding the percentage for unmarried cases.

Response: We have added the percentage for unmarried cases.

 It is 53.7% (Line 268).

6. Line 261 and 290, add .0 to the 82% and 98%

Response: We corrected the text (Line 272 and Lines 301-303) .

7. Line 272, IQR should be only a number as it presents de distance between quartiles 25th and 75th. If the authors are expressing the quartiles 25 and 75, please correct it.

Response: We corrected the text (Line 283).

8. Line 291, I am not sure what this 25% inside the parenthesis stands for.

Response: We corrected the text: “When asked about the type of affiliation to the social security system in Colombia, it was found that 25.5% were affiliated to the contributory system (n: 11,449)” (Lines 301-303)

9. Line 437 and 440, Consider adding a full stop instead of a semicolon before “in consistency” and “therefore”, respectively.

Response: We corrected the text. (Line 451 and Line 454)

10. References, I encourage the authors to carefully check all the references.

Response: We have carefully reviewed the references and made the necessary adjustments.

11. I would remove the PMID if already giving the doi.

Response:We made the requested adjustments (References 25 (Line 769) and 26 (Line 773)) 

12. Reference 36, Injury Prevention is the name of the journal I am not sure what does 6.4 stands for, please check it and correct it if necessary.

Response:We corrected the text (Reference 40 (Line 824))

13. Reference 39, “Ciênc. saúde colet” should be each word capitalized as it is the name of the journal. Same for reference 28.

Response: We made the requested adjustments (Line 836 and Line 789) 

Academic editor review

Personally, and after a careful read of your revised paper, I would also like to ask you to strengthen the literature review, specifically in regards to the empirical evidences on human factors and road safety, e.g., risky road behaviors in the context of Medellín, or at least Colombian/Latin American cities, and the relationship between the specific road environment of the country and toad behaviors and crashes. This is especially problematic if the fact that most of the papers empirically documenting this issue were not considered in your manuscript, and many of the study assumptions were done on the basis of non-scientific sources. In other words, in its present form, the paper lacks from a clear scientific support on the road safety panorama of Colombian drivers (including motorcyclists).

Response to academic editor

We consider that our study has a very clear aim, which is the estimation of the burden of disease due to road incidents of motorcyclists, and does not focus on the factors that have an impact on its occurrence, such as the human factor. However, we decided to accept your suggestion, given the importance of the relationship between the human factor and road safety, and therefore we annexed to the manuscript, in some sections, especially in the one related to practical implications, aspects that relate the human factor and road safety. These are indicated below (the line numbers serve as an orientation for the immediate location of the text in the manuscript with no change control):

There are other endogenous factors to be taken into account in RIs, especially the human factor. In Colombia, Norza et al. in 2014, in a sample of 16,322 drivers, reported the role of motorcyclists in road incidents, probably attributed to the disregard of traffic regulations, which added the fact of the lack experience behind the wheel that characterizes these people. The long working hours combined with the drivers' lack of emotional intelligence could be triggers of stress, anxiety, anger, aggressiveness, hostility, dissociative disorders, and consequently, factors that facilitate the tendency to commit more infractions and RIs (1). Also, in three Colombian cities, Ibague-Valledupar, and Bogotá, two studies ratified the precept of the multi-causality of traffic accidents, where human factors are the ones that most contributed to the occurrence of incidents, aggravated by apparent deficiencies in infrastructure and some organizational problems (2-3) 

Lines 66-77

In this regard, other factors that have been related to the higher death rates due to RIs, in general, in drivers under 25 years of age, are the lack of experience, unsafe attitudes, sensation seeking, lower risk perception and lower use of safety measures (4)

Lines 462-465

The combination of three factors, human, environmental, and automotive, constitute the fundamental elements to be considered in road events. In particular, knowledge of the human factor is essential in road safety to guide intervention strategies and prevent accidents and their causes. The human factor is the most difficult to control and the most influential because driving skills and habits are decisive for accident prevention. Although in theory, drivers, in general, should be in optimal psychophysical conditions when driving, in practice, this is not usual, and therefore deficiencies in psychophysical health can trigger events that can life-threatening to drivers, passengers, and pedestrians. In particular, studies addressing the human factor analysis in road incidents involving motorcyclists are scarce, probably due to the complexity of the evaluation of human characteristics.

Lines 594-604

Some studies have focused on the human factor in motorcycle drivers involved in RIs in Latin American countries. A descriptive study in Paraguay for this specific population concluded that the greatest attribution to the occurrence of road events was due to behavioral factors (5). In Argentina, the costs and factors derived from motorcycle accidents were analyzed, highlighting that the human factor was the leading cause, especially the lack of respect for traffic regulations, within which speeding when performing maneuvers can cause harm to the driver and passengers (6). In Caracas, in a case-control design, a positive association was found between the presence of Attention Deficit Hyperactivity Disorder (ADHD) and a higher occurrence RIs of motorcyclist in people between 18 and 63 years of age, i.e., motorcycle drivers who presented the disease had a higher risk of experiencing RIs incidents due to their greater inclination to violate traffic laws and to have inappropriate behaviors than motorcycle drivers who did not present ADHD (7). 

Lines 605-618

In Medellin in 2015 (8) , the psychological profile of drivers who repeatedly violated traffic rules were analyzed, identifying that reckless, risky, and aggressive driving, disrespect for traffic rules, emotional instability, especially anxiety in young drivers, and dissociative personality were related to attentional errors in driving and traffic incidents. The greater impact on road events in the young population could be explained by their perception of risk, which is also affected by perceptual abilities, driving experience, emotional state, and traffic regulations (9). 

Lines 619-625

In particular, the trait of road rage, as a human factor, has been little studied in the context of professional drivers. Useche et al., in 2019 (10), analyzed the association between trait road rage and driving styles in a sample of Colombian professional drivers, highlighting, among the main findings that impeding forward progress by other drivers, illegal driving and hostility significantly predicted adaptive and maladaptive driving styles. It was concluded that interventions that focus on minimizing the driving anger profile of drivers could be effective in counteracting the consequences of such predisposition. Apart from the behavioral problems referred to above, motorcycle drivers in the city of Medellin are known for their special predisposition to contravene road regulations. In a comparative study on the profile of motorcyclists in four Colombian cities, Bogotá, Medellin, Cucuta, and Bucaramanga (11), it was found that it is in Medellin where this type of driver violates the National Traffic Code to a greater extent, and it was concluded that the motorcyclist‘s profile is different depending on the region they are from and therefore it is ratified that generalized public policies cannot be carried out by local governments towards issues related to motorcycles.

Lines 627-642

The human factor is the most difficult to control and is influential since driving skills and habits determine accident prevention. In this sense, behavioral interventions are essential for structuring safe road systems and, therefore, require the design of intervention programs to change driver behavior (12). Necessarily, in the perspective of minimizing the risks inherent to the human factor, prior knowledge of the causal factors is required through antecedent and consequent communication strategies with the driver and the application of principles of social influence, which allow creating suitable conditions for the desired behavior to occur.

References

(1) Norza-Céspedes EH, Granados-León EL, Useche-Hernández SA, Romero-Hernández M, Moreno-Rodríguez J. Descriptive and explanatory components of road accident rate in Colombia: influence of the human factor. Revista criminalidad. 2014; 56(1):157-187.Available: http://www.scielo.org.co/scielo.php?script=sci_arttext&pid=S1794-31082014000100009) 

(2) Flórez CF, Patiño C, Rodríguez JM, Ariza LK, LIANY; González RA. Análisis multicausal de ‘accidentes’ de tránsito en dos ciudades de Colombia. Archivos de Medicina (Manizales). 2018; 18(1); 69-85. doi:: https://doi.org/10.30554/archmed.18.1.2477.2018

(3)Castellanos OJ. Análisis de la tasa de accidentalidad de motocicletas entre los años 2005–2017 en la ciudad de Bogotá DC.Universidad Cooperativa de Colombia. 2018. Available: https://repository.ucc.edu.co/bitstream/20.500.12494/10185/1/2018_%20Analisis_Accidentalidad_Motocicletas.pdf

(4) Useche S. A., Alonso F, Montoro, L, Esteban C. Explaining self-reported traffic crashes of cyclists: An empirical study based on age and road risky behaviors. Safety science. 2019; 113: 105-114. doi: https://doi.org/10.1016/j.ssci.2018.11.021

(5) Negret MM, Sequera G. Factores sociales, ambientales y comportamentales relacionados con los siniestros de tránsito que involucran motocicletas en Asunción, Paraguay. Revista Científica OMNES. 2020; 3(2): 79-109. Available: https://www.columbia.edu.py/investigacion/ojs/index.php/OMNESUCPY/article/view/72

(6) Besse M, Denari R, Villani A, San Roque M, Rosado J, Sarotto AJ. Accidentes de moto: costo médico/económico en un hospital municipal de la ciudad de Buenos Aires. MEDICINA (Buenos Aires). 2018; 78(3):158-162. Available: https://www.medicinabuenosaires.com/revistas/vol78-18/n3/158-162-Med6767-Besse.pdf

(7) Suárez I, Rosales-Pereira KA, Silva AA, González G, López K, Rísquez A. Déficit de Atención e Hiperactividad. Su relación con Accidentes de Tránsito en Motocicletas. Medicina Interna. 2019; 35(3): 99-106. Available: https://www.svmi.web.ve/ojs/index.php/medint/article/view/520

(8) Durán-Palacio NM, Moreno-Carmona ND. Personalidad e infracciones frecuentes de normas de tránsito. Diversitas. 2016; 12(1): 123-136. doi: https://doi.org/10.15332/s1794-9998.2016.0001.09

(9) Payani S, Hamid H, Law TH. A review on impact of human factors on road safety with special focus on hazard perception and risk-taking among young drivers. In IOP Conference Series: Earth and Environmental Science. 2019; 357(1). Available: https://iopscience.iop.org/article/10.1088/1755-1315/357/1/012041/meta. 

(10) Useche SA, Cendales B, Alonso F, Montoro L, Pastor JC. Trait driving anger and driving styles among Colombian professional drivers. Heliyon. 2019); 5(8). doi: https://doi.org/10.1016/j.heliyon.2019.e02259

(11) (Martin-Rojas DA, Pardo-Castaño, D. Estudio comparativo del perfil del motociclista en cuatro ciudades de Colombia: Bogotá, Medellín, Cúcuta y Bucaramanga. Ciencia Unisalle. 2019. Available: https://ciencia.lasalle.edu.co/ing_civil/535 ) 

(12) Theeuwes J, van der Horst R, Kuiken M. Designing safe road systems : a human factors perspective. Jan Series: Human factors in road and rail transport Publisher: CRC Press. 2017

---

## [Editor Report · Decision Letter 2]

16 Aug 2021

Loss of years of healthy life due to road incidents of motorcyclists in the city of Medellin, 2012 to 2015

PONE-D-20-40988R2

Dear Dr. Grisales-Romero,

We’re pleased to inform you that your manuscript has been judged scientifically suitable for publication and will be formally accepted for publication once it meets all outstanding technical requirements.

Kind regards,

Sergio A. Useche, Ph.D.

Academic Editor

PLOS ONE

Additional Editor Comments (optional): The manuscript is much better now. The authors have done a good job in addressing all the reviewers' and editor's queries, and can be published in its current form. Good job!

---

## [Editor Report · Acceptance letter]

19 Aug 2021

PONE-D-20-40988R2 

Loss of Years of Healthy Life Due to Road Incidents of Motorcyclists in the City of Medellin, 2012 to 2015 

Dear Dr. Grisales-Romero:

I'm pleased to inform you that your manuscript has been deemed suitable for publication in PLOS ONE. Congratulations! Your manuscript is now with our production department. 

Kind regards, 

on behalf of

Dr. Sergio A. Useche 

Academic Editor

PLOS ONE